# A Simple Open-Loop Baseline for Reinforcement Learning Locomotion Tasks

## Abstract

In search of the simplest baseline capable of competing with Deep Reinforcement Learning on locomotion tasks, we propose a biologically inspired model-free open-loop strategy. Drawing upon prior knowledge and harnessing the elegance of simple oscillators to generate periodic joint motions, it achieves respectable performance in five different locomotion environments, with a number of tunable parameters that is a tiny fraction of the thousands typically required by RL algorithms. Unlike RL methods, which are prone to performance degradation when exposed to sensor noise or failure, our open-loop oscillators exhibit remarkable robustness due to their lack of reliance on sensors. Furthermore, we showcase a successful transfer from simulation to reality using an elastic quadruped, all without the need for randomization or reward engineering. Overall, the proposed baseline and associated experiments highlight the existing limitations of DRL for robotic applications, provide insights on how to address them, and encourage reflection on the costs of complexity and generality.

## 1 Introduction

The field of deep reinforcement learning (DRL) has witnessed remarkable strides in recent years, pushing the boundaries of robotic control to new frontiers (Song et al., 2021; Hwangbo et al., 2019). However, a dominant trend in the field is the steady escalation of algorithmic complexity. As a result, the latest algorithms require a multitude of implementation details to achieve satisfactory performance levels (Huang et al., 2022), leading to a concerning reproducibility crisis (Henderson et al., 2018). Moreover, even state-of-the-art DRL models struggle with seemingly simple problems, such as the Mountain Car environment (Colas et al., 2018) or the Swimmer task (Franceschetti et al., 2022; Huang et al., 2023).

Fortunately, several works have gone against the prevailing direction and tried to find simpler baselines, scalable alternatives for RL tasks (Rajeswaran et al., 2017; Salimans et al., 2017; Mania et al., 2018). These efforts have not only raised questions about the evaluation and trends in RL (Agarwal et al., 2021), but also highlighted the need for simplicity in the field. In this paper, we carry this torch further by introducing an extremely simple open-loop trajectory generator that operates independently of sensor data.

While the exploration of policy structures in deep RL algorithms has remained relatively under-explored, our research underscores its significance. We demonstrate that, by adopting the right structural elements, even a minimal policy can achieve satisfactory performance levels[1]. The generality of RL algorithms is undeniable, but it comes at the price of specificity in task design, in the form of complex reward engineering (Lee et al., 2020). We advocate leveraging prior knowledge to reduce complexity, both in the algorithm and in the task formulation, when tackling specific problem categories such as locomotion tasks. In particular, we show that simple open-loop oscillators can provide an effective and efficient solution for locomotion challenges. The proposed open-loop approach not only reduces the computational load and reward engineering effort, but also facilitates the deployment of policies on real embedded systems, making it a valuable asset for practical applications.

---

[1]See the 35 lines of code to solve Swimmer in Fig. 6 of Appendix A.1

The open-loop strategy we present offers a profound advantage – its resilience in the face of the inherent noise and unpredictability of sensory inputs (Goodwin et al., 2000; Ijspeert, 2008). Unlike conventional reinforcement learning, which is notoriously brittle in the presence of sensor noise or changes in the environment (Liu et al., 2023), our approach remains consistent in its performance. We argue that this robustness is valuable for real-world applications, where perfect information and unchanging conditions are elusive ideals.

Our intention is not to replace DRL algorithms, as the open-loop strategy has clear limitations and cannot compete in complex locomotion scenarios. Rather, our goal is to highlight the existing limitations of DRL, provide insights, and encourage reflection on the costs of complexity and generality. This is achieved by studying one of the simplest model-free open-loop baseline.

## 1.1 CONTRIBUTIONS

In summary, the main contributions of our paper are:

- a simple open-loop baseline for learning locomotion that can handle sparse rewards and high sensory noise and that requires very few parameters (on the order of tens, Section 2),
- showing the importance of prior knowledge and choosing the right policy structure (Section 4.2),
- a study of the robustness of RL algorithms to noise and sensor failure (Section 4.3),
- showing successful simulation to reality transfer, without any randomization or reward engineering, where deep RL algorithms fail (Section 4.4).

## 2 OPEN-LOOP OSCILLATORS FOR LOCOMOTION

We draw inspiration from nature and specifically from central pattern generators, as explored by Righetti et al. (2006); Raffin et al. (2022); Bellegarda & Ijspeert (2022). Our approach leverages nonlinear oscillators with phase-dependent frequencies to produce the desired motions for each actuator. The equation of one oscillator is:

$$q_i^{\text{des}}(t) = a_i \cdot \sin(\theta_i(t) + \varphi_i) + b_i$$
$$\dot{\theta}_i(t) = \begin{cases} \omega_{\text{swing}} & \text{if } \sin(\theta_i(t) + \varphi_i) > 0 \\ \omega_{\text{stance}} & \text{otherwise} \end{cases} \tag{1}$$

where $q_i^{\text{des}}$ is the desired position for the i-th joint, $a_i$, $\theta_i$, $\varphi_i$ and $b_i$ are the amplitude, phase, phase shift and offset of oscillator $i$. $\omega_{\text{swing}}$ and $\omega_{\text{stance}}$ are the frequencies of oscillations in rad/s for the swing and stance phases. To keep the search space small, we use the same frequencies $\omega_{\text{swing}}$ and $\omega_{\text{stance}}$ for all actuators.

This formulation is both simple and fast to compute; in fact, since we do not integrate any feedback term, all the desired positions can be computed in advance. The phase shift $\varphi_i$ plays the role of the coupling term found in previous work: joints that share the same phase shift oscillate synchronously. However, compared to previous studies, the phase shift is not pre-defined but learned.

Optimizing the parameters of the oscillators is achieved using black-box optimization (BBO), specifically the CMA-ES algorithm (Hansen et al., 2003; Hansen, 2009) implemented within the Optuna library (Akiba et al., 2019). This choice stems from its performance in our initial studies and its ability to escape local minima. In addition, because BBO uses only episodic returns rather than immediate rewards, it makes the baseline robust to sparse or delayed rewards. Finally, a proportional-derivative (PD) controller converts the desired joint positions generated by the oscillators into desired torques.

## 3 RELATED WORK

**The quest for simpler RL baselines.** Despite the prevailing trend towards increasing complexity, some research has been dedicated to developing simple yet effective baselines for solving robotic tasks using RL. In this vein, Rajeswaran et al. (2017) proposed the use of policies with simple

parametrization, such as linear or radial basis functions (RBF), and highlighted the brittleness of RL agents. Concurrently, Salimans et al. (2017) explored the use of evolution strategies as an alternative to RL, exploiting their fast runtime to scale up the search process. More recently, Mania et al. (2018) introduced Augmented Random Search (ARS), a straightforward population-based algorithm that trains linear policies. Building upon these efforts, we seek to further simplify the solution by proposing open-loop oscillators to generate desired joint trajectories independently of the robot's state. Our goal is to provide the simplest model-free method capable of achieving respectable performance on standard locomotion tasks.

**Biology inspired locomotion.** Biological studies have extensively investigated the role of oscillators as fundamental components of locomotion (Delcomyn, 1980; Cohen & Wallén, 1980; Ijspeert, 2008), including the identification of central pattern generators (CPGs) – neural networks capable of generating synchronized patterns of activity, without relying on rhythmic input from sensory feedback – in animals such as lampreys (Ijspeert, 2008). Inspired by these findings, researchers have incorporated oscillators into robotic control for locomotion (Crespi & Ijspeert, 2008; Iscen et al., 2013), and recent works have combined learning approaches with CPGs in task space for quadruped locomotion (Kohl & Stone, 2004; Tan et al., 2018; Iscen et al., 2018; Yang et al., 2022; Bellegarda & Ijspeert, 2022; Raffin et al., 2022). However, surprisingly and to the best of our knowledge, no previous studies have explored the use of open-loop oscillators in reinforcement learning locomotion benchmarks, possibly due to the belief that open-loop control is insufficient for stable locomotion (Iscen et al., 2018). Our work aims to address this gap by evaluating simple open-loop oscillators in RL locomotion tasks and on a real hardware, directly in joint space, eliminating the need for inverse kinematics and pre-defined gaits.

## 4 RESULTS

We assess the effectiveness of our method through experiments on locomotion tasks across diverse environments, including simulated tasks and transfer to a real elastic quadruped.

Our goal is to address three key questions:

- How do simple open-loop oscillators fare against deep reinforcement learning methods in terms of performance, runtime and parameter efficiency?
- How resilient are RL policies to sensor noise, failures and external perturbations when compared to the open-loop baseline?
- How do learned policies transfer to a real robot when training without randomization or reward engineering?

By examining these questions, we seek to provide a comprehensive understanding of the strengths and limitations of our proposed approach and shed light on the potential benefits of leveraging prior knowledge in robotic control.

### 4.1 IMPLEMENTATION DETAILS

For the RL baselines, we utilize JAX implementations from Stable-Baselines3 (Bradbury et al., 2018; Raffin et al., 2021a) and the RL Zoo (Raffin, 2020) training framework. The search space used to optimize the parameters of the oscillators is shown in Table 3 of Appendix A.2.

### 4.2 RESULTS ON THE MUJOCO LOCOMOTION TASKS

We assess the efficacy of our method on the MuJoCo v4 locomotion tasks (ANT, HALFCHEE-TAH, HOPPER, WALKER2D, SWIMMER) included in the Gymnasium v0.29.1 library (Towers et al., 2023). We compare our approach against three established deep RL algorithms: Proximal Policy Optimization (PPO), Deep Deterministic Policy Gradients (DDPG), and Soft Actor-Critic (SAC). To ensure a fair comparison, we adopt the hyperparameter settings from the original papers, except for the swimmer task, where we fine-tuned the discount factor ($\gamma = 0.9999$) according to Franceschetti et al. (2022). Additionally, we also benchmark Augmented Random Search (ARS) which is a population based algorithm that uses linear policies. Our choice of baselines includes one representative

example per algorithm category: PPO for on-policy, SAC for off-policy, ARS for population-based methods and simple model-free baselines, and DDPG as a historical algorithm (many state-of-the-art algorithms are based on it). We choose SAC (Haarnoja et al., 2019) because it performs well in continuous control tasks (Huang et al., 2023), and it shares many components (including the policy structure) with its newer and more complex variants. SAC and its variants, such as TQC (Kuznetsov et al., 2020), REDQ (Chen et al., 2021) or DroQ (Hiraoka et al., 2022) are also the ones used in the robotics community (Raffin et al., 2022; Smith et al., 2023). We use standard reward functions provided by Gymnasium, except for ARS where we remove the alive bonus to match the results from the original paper.

The RL agents are trained during one million steps. To have quantitative results, we replicate each experiment 10 times with distinct random seeds. We follow the recommendations by Agarwal et al. (2021) and report performances profiles, probability of improvements in Fig. 1 and aggregated metrics with 95% confidence intervals in Fig. 2. We normalize the score over all environments using a random policy for the minimum and the maximum performance of the open-loop oscillators.

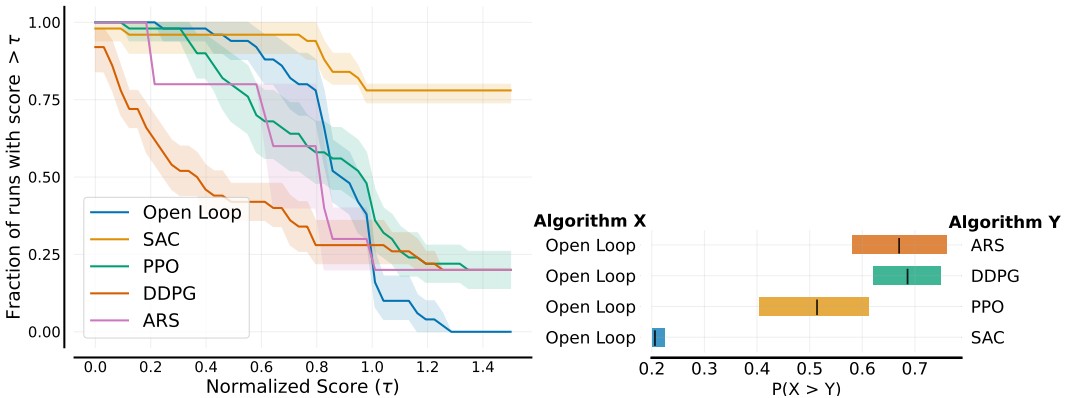

Figure 1: Performance profiles on the MuJoCo locomotion tasks (left) and probability of improvements of the open-loop approach over baselines, with a 95% confidence interval.

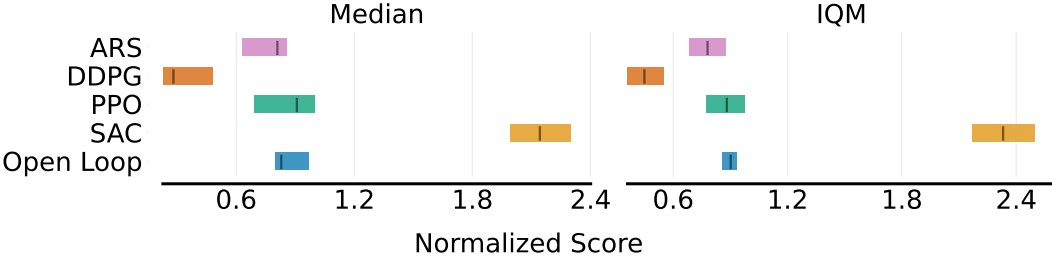

Figure 2: Metrics results on MuJoCo locomotion tasks using median and interquartile mean (IQM), with a 95% confidence interval.

**Performance.** As seen in Figs. 1 and 2, the open-loop oscillators achieves respectable performance across all five tasks, despite its minimalist design. In particular, it performs favorably against ARS and DDPG, a simple baseline and a classic deep RL algorithm, and exhibits comparable performance to PPO. Remarkably, this is accomplished with merely a dozen parameters, in contrast to the thousands typically required by deep RL algorithms. Our results suggest that simple oscillators can effectively compete with sophisticated RL methods for locomotion, and do so in an open-loop fashion. It also shows the limits of the open-loop approach. The baseline does not reach the maximum performance of SAC.

**Runtime.** Comparing the runtime of the different algorithms[2], as presented in Table 1, underscores the benefits of choosing simplicity over complexity. Notably, ARS requires only five minutes of CPU

---

[2]We display the runtime for HALFCHEETAH only, the computation time for the other tasks is similar.

Table 1: Runtime comparison to train a policy on HALFCHEETAH, one million steps using a single environment, no parallelization.

| | SAC | | PPO | | DDPG | | ARS | | Open-Loop | |
|---|---|---|---|---|---|---|---|---|---|---|
| | CPU | GPU | CPU | GPU | CPU | GPU | CPU | GPU | CPU | GPU |
| Runtime (in min.) | 80 | 30 | 10 | 14 | 60 | 25 | 5 | N/A | **2** | N/A |

time to train on a single environment for one million steps, while open-loop oscillators are twice as fast. This efficiency becomes particularly advantageous when deploying policies on embedded systems with limited computing resources. Moreover, both methods can be easily scaled using asynchronous parallelization to achieve satisfactory performance in a timely manner. In contrast, more complex methods like SAC demand a GPU to achieve reasonable runtimes (15 times slower than open-loop oscillators), even with the aid of JIT compilation[3].

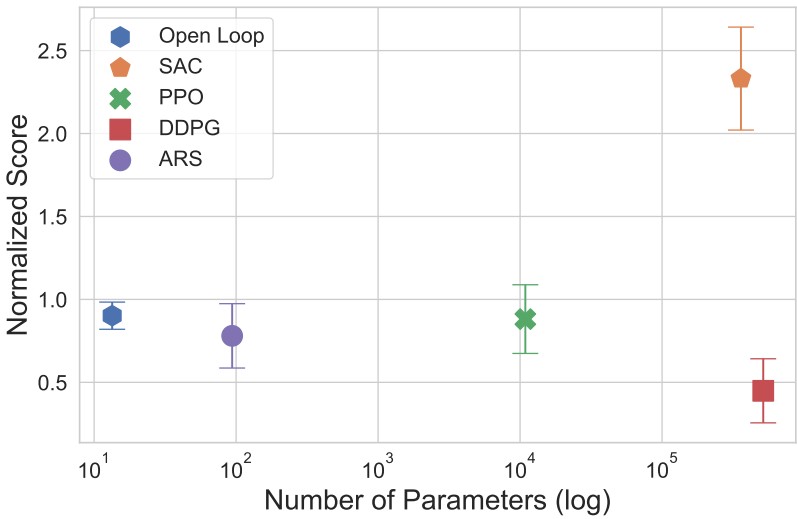

Figure 3: Parameter efficiency of the different algorithms. Results are presented with a 95% confidence interval and score are normalized with respect to the open-loop baseline.

**Parameter efficiency.** As seen in Fig. 3, the open-loop oscillators really stand out for their simplicity and performance with respect to the number of optimized parameters. On average, our approach has 7x fewer parameters than ARS, 800x fewer than PPO and 27000x fewer than SAC. This comparison highlights the importance of choosing an appropriate policy structure that delivers satisfactory performance while minimizing complexity.

### 4.3 ROBUSTNESS TO SENSOR NOISE AND FAILURES

In this section, we assess the resilience of the trained agents from the previous section against sensor noise, malfunctions and external perturbations (Dulac-Arnold et al., 2020; Seyde et al., 2021). To study the impact of noisy sensors, we introduce Gaussian noise with varying intensities into one sensor signal (specifically, the first index in the observation vector, the one that gives the position of the end-effector). To investigate the robustness against sensor faults, we simulate two types of sensor failures: Type I failure involves outputting zero values for one sensor, while Type II failure generates a constant value with a larger magnitude (we set this value to five in our experiments). Finally, we evaluate the robustness to external disturbances by applying perturbations with a force of 5N in randomly chosen directions with a probability of 5% (around 50 impulses per episode). By examining how the agents perform under these scenarios, we can evaluate their ability to adapt to imperfect sensory input and react to disturbances. We study the effect of randomization by also

---

[3]The JAX implementation of SAC used in this study is four times faster than its PyTorch counterpart.

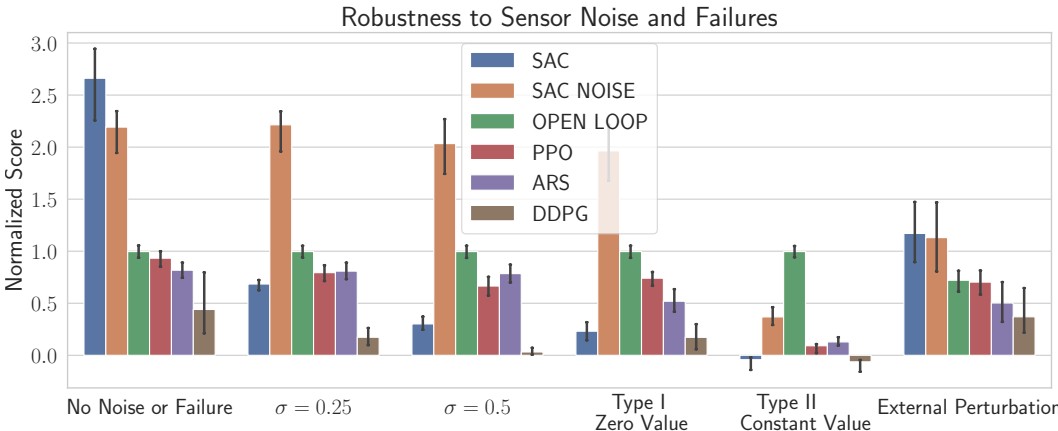

Figure 4: Robustness to sensor noise (with varying intensities), failures of Type I (all zeros) and II (constant large value) and external disturbances. All results are presented with a 95% confidence interval and score are normalized with respect to the open-loop baseline.

training SAC with a Gaussian noise with intensity $\sigma = 0.2$ on the first sensor (SAC NOISE in the figure).

In absence of noise or failures, SAC excels over simple oscillators on most tasks, except for the SWIMMER environment. However, as depicted in Fig. 4, SAC performance deteriorates rapidly when exposed to noise or sensor malfunction. This is the case for the other RL algorithms, where ARS and PPO are the most robust ones but still exhibit degraded performances. In contrast, open-loop oscillators remain unaffected, except when exposed to external perturbations because they do not rely on sensors. This highlights one of the primary advantages and limitations of open-loop control.

As shown by the performance of SAC trained with noise on the first sensor (SAC NOISE), it is possible to mitigate the impact of sensor noise. This result, together with the performance of the open-loop controller, suggests that the first sensor is not essential for achieving good performance in the MuJoCo locomotion tasks. SAC with randomization on the first sensor has learned to disregard its input, while SAC without randomization exhibits a high sensitivity to the value of this uninformative sensor. This finding illustrates a vulnerability of DRL algorithms, which can be sensitive to useless inputs.

## 4.4 SIMULATION TO REALITY TRANSFER ON AN ELASTIC QUADRUPED

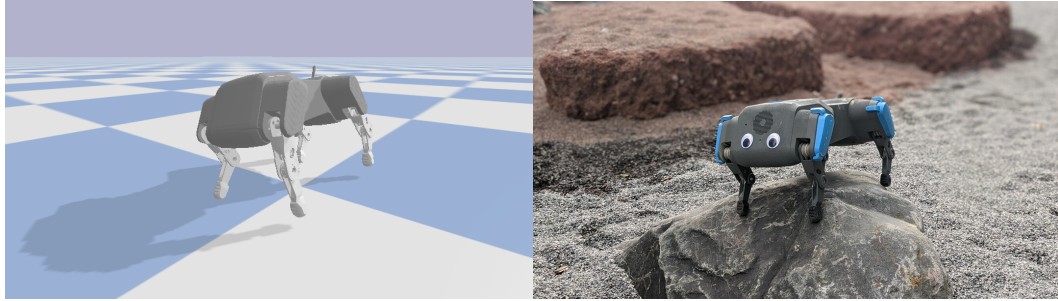

Figure 5: Robotic quadruped with elastic actuators in simulation (left) and real hardware (right)

The open-loop approach offers a promising solution for locomotion control on real robots, owing to its computational efficiency, resistance to sensor noise, and adequate performance. To assess

its potential for real-world applications, we investigate whether the results in simulation can be transferred to a real quadruped robot equipped with serial elastic actuators[4].

The experimental platform is a cat-sized quadruped robot with eight joints, similar to the ANT task in MuJoCo, where motors are connected to the links via a linear torsional spring with constant stiffness $k \approx 2.75$Nm/rad. To conduct our evaluation, we utilize a simulation of the robot in PyBullet (Coumans & Bai, 2016–2021), which includes a model of the elastic joints but excludes motor dynamics. The task is to reach maximum forward speed: we define the reward as displacement along the desired axis and limit each episode to five seconds of interaction. The agent receives the current joint positions $q$ and velocities $\dot{q}$ as observation and commands desired joint positions $q^{\text{des}}$ at a rate of 60Hz.

In this evaluation, we compare the open-loop approach against the top-performing algorithm from Section 4.2, namely SAC. Both algorithms are allotted a budget of one million steps for training. Importantly, we do not apply any randomization or task-specific techniques during the training process. Our goal is to understand the strengths and weaknesses of RL with respect to the open-loop baseline in a simple simulation-to-reality setting. We evaluate the learned policy from simulation on the real robot for ten episodes.

Table 2: Results of simulation-to-reality transfer for the elastic quadruped locomotion task. We report mean speed and standard error over ten test episodes. SAC performs well in simulation, but fails to transfer to the real world.

|  | SAC | | Open-Loop | |
| --- | --- | --- | --- | --- |
|  | Sim | Real | Sim | Real |
| Mean speed (m/s) | **0.81** +/ 0.02 | 0.04 +/ 0.01 | 0.55 +/ 0.03 | **0.36** +/ 0.01 |

As shown in Table 2, SAC exhibits superior performance in simulation compared to the open-loop oscillators (like in Section 4.2), with a mean speed of 0.81 m/s versus 0.55 m/s over ten runs. However, upon closer examination, the policy learned by SAC outputs high-frequency commands making it unlikely to transfer to the real robot – a common issue faced by RL algorithms (Raffin et al., 2021b; Bellegarda & Ijspeert, 2022). When deployed on the real robot, the jerky motion patterns translate into suboptimal performance (0.04 m/s), commands that can damage the motors, and increased wear-and-tear.

In contrast, our open-loop oscillators, with fewer than 25 adjustable parameters, produce smooth outputs by design and demonstrate good performance on the real robot. The open-loop policy achieves a mean speed of 0.36 m/s, the fastest walking gait recorded for this elastic quadruped (Lakatos et al., 2018). While there is still a disparity between simulation and reality, the gap is significantly narrower compared to the RL algorithm.

## 5 DISCUSSION

**A simple open-loop model-free baseline.** We propose a simple, open-loop model-free baseline that achieves satisfactory performance on standard locomotion tasks without requiring complex models or extensive computational resources. While it does not outperform RL algorithms in simulation, this approach has several advantages for real-world applications, including fast computation, ease of deployment on embedded systems, smooth control outputs, and robustness to sensor noise. These features help narrow the simulation-to-reality gap and avoid common issues associated with deep RL algorithms, such as jerky motion patterns (Raffin et al., 2021b) or converging to a bang-bang controller (Seyde et al., 2021). Our approach is specifically tailored to address locomotion tasks, yet its simplicity does not limit its versatility. It can successfully tackle a wide array of locomotion challenges and transfer to a real robot, with just a few tunable parameters, while remaining model-free.

**The cost of generality.** Deep RL algorithms for continuous control often strive for generality by employing a versatile neural network architecture as the policy. However, this pursuit of generality

---

[4]The results can also be seen in the video in the supplementary material.

comes at a price of specificity in the task design. Indeed, the reward function and action space must be carefully crafted to solve the locomotion task and avoid solutions that hack the simulator but do not transfer to the real hardware. Our study and other recent work (Iscen et al., 2018; Bellegarda & Ijspeert, 2022; Raffin et al., 2022) suggest incorporating domain knowledge into the policy design. Even minimal knowledge like simple oscillators, reduces the search space and the need for complex algorithms or reward design.

**RL for more complex locomotion scenarios.** The locomotion tasks presented in this paper may seem relatively simple compared to the more complex challenges that RL has tackled (Miki et al., 2022). However, the MuJoCo environments have served as a benchmark for the continuous control algorithms used on robots and are still widely utilized in both online and offline RL. It is important to note that even SAC, which performs well in simulation, can perform sub-optimally with simple environments like the swimmer task (Franceschetti et al., 2022) or the elastic quadruped simulation-to-reality transfer, and be sensitive to uninformative sensors. We believe that understanding the failures and limitations by providing a simple open-loop model-free baseline is more valuable than marginally improving performance by adding new tricks to an already complex algorithm (Patterson et al., 2023).

**Unexpected results.** While the success of the open-loop oscillators in the SWIMMER environment is anticipated, their effectiveness in the WALKER, HOPPER or elastic quadruped environments is more unexpected, as one might assume that feedback control or inverse kinematics would be necessary to balance the robots or to learn a meaningful open-loop policy. While it is true that previous studies have shown that periodic control is at the heart of locomotion (Ijspeert, 2008), we argue that the required periodic motion can be surprisingly simple. Mania et al. (2018) have shown that simple linear policies can be used for locomotion tasks. The present work goes a step further by reducing the number of parameters by a factor of ten and removing the state as an input.

**Exploiting robot natural dynamics.** Our open-loop baseline reveals an intriguing insight: a single frequency per phase (swing or stance) can be employed across all joints for all considered tasks. This observation resonates with recent research focused on exploiting the natural dynamics of robots, particularly using nonlinear modes that enable periodic motions with minimal actuation (Della Santina et al., 2020; Albu-Schäffer & Della Santina, 2020; Albu-Schäffer & Sachtler, 2022). Our approach could potentially identify periodic motions for locomotion while minimizing control effort, thus harnessing the inherent dynamics of the hardware.

**Limitations** Naturally, open-loop control alone is not a complete solution for locomotion challenges. Indeed, by design, open-loop control is vulnerable to disturbances and cannot recover from potential falls. In such cases, closing the loop with reinforcement learning becomes essential to adapt to changing conditions, maintain stability or follow a desired goal. A hybrid approach that integrates the strengths of feedforward (open-loop) and feedback (closed-loop) control offers a middle ground, as seen in various engineering domains (Goodwin et al., 2000; Astrom & Murray, 2008; Della Santina et al., 2017). By combining the speed and noise resilience of open-loop control with the adaptability of closed-loop control, it enables reactive and goal-oriented locomotion. Prior studies have explored this combination (Iscen et al., 2018; Bellegarda & Ijspeert, 2022; Raffin et al., 2022), but our research simplifies the feedforward formulation and eliminates the need for inverse kinematics or predefined gaits.

**Future work.** While our approach generates desired joint positions using oscillators without relying on the robot state, a PD controller is still required in simulation to convert these positions into torque commands. We consider this requirement as part of the environment, since a position interface is usually provided when considering real robotic applications. Furthermore, the generated torques appear to be periodic, suggesting that the PD controller could be replaced by additional oscillators (additional harmonic terms). While this possibility is worth exploring, we focus on simplicity in our current work, using a minimal number of parameters, and defer this endeavor to future research.

REPRODUCIBILITY STATEMENT

We provide a minimal standalone code (35 lines of Python code) in the Appendix (Fig. 6) that allows to solve the SWIMMER task using open-loop oscillators. The code to reproduce the main experiments is provided in the supplementary material. The search space and details for optimizing the oscillators parameters are given in Appendix A.2.

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

## A APPENDIX

### A.1 STANDALONE CODE FOR THE SWIMMER TASK

```python
import gymnasium as gym
import numpy as np
from gymnasium.envs.mujoco.mujoco_env import MujocoEnv

# Env initialization
env = gym.make("Swimmer-v4", render_mode="human")
# Wrap to have reward statistics
env = gym.wrappers.RecordEpisodeStatistics(env)
mujoco_env = env.unwrapped
n_joints = 2
assert isinstance(mujoco_env, MujocoEnv)
# PD Controller gains
kp, kd = 10, 0.5
# Reset the environment
t, _ = 0.0, env.reset(seed=0)
# Oscillators parameters
omega = 2 * np.pi * 0.62 * np.ones(n_joints)
phase = 2 * np.pi * np.array([0.00, 0.95])

while True:
    env.render()
    # Open-Loop Control using oscillators
    desired_qpos = np.sin(omega * t + phase)
    # PD Control: convert to torque, desired qvel is zero
    desired_torques = (
        kp * (desired_qpos - mujoco_env.data.qpos[-n_joints:])
        - kd * mujoco_env.data.qvel[-n_joints:]
    )
    desired_torques = np.clip(desired_torques, -1.0, 1.0)  # clip to action bounds
    _, reward, terminated, truncated, info = env.step(desired_torques)
    t += mujoco_env.dt

    if terminated or truncated:
        print(f"Episode return: {float(info['episode']['r']):.2f}")
        t, _ = 0.0, env.reset()
```

Figure 6: Minimal code to solve the SWIMMER environment using open-loop oscillators (highlighted in black). Code was tested with Gymnasium v0.29.1, MuJoCo v2.3.7 and Python 3.9.

## A.2 OPEN-LOOP OSCILLATORS SEARCH SPACE

Table 3: Search space for the oscillators parameters. We set $\varphi_0 = 0$ by convention, use a step-size $dt = 0.001$ for the integration of the oscillators equations and have a population size of 30 for CMAES. $\mathcal{U}(-1, 1)$ means that the value is sampled from a uniform distribution between $-1$ and $1$. For the SWIMMER task, a constant amplitude and offset are used.

|                | Amplitude $a_i$  | Offset $b_i$     | Phase Shift $\varphi_i$     | Frequencies $\omega_{\text{swing/stance}}$ |
|----------------|------------------|------------------|-----------------------------|--------------------------------------------|
| Ant-v4         | $\mathcal{U}(-1, 1)$ | $\mathcal{U}(-1, 1)$ | $2\pi \cdot \mathcal{U}(0, 1)$ | $2\pi \cdot \mathcal{U}(0.4, 2)$           |
| HalfCheetah-v4 | $\mathcal{U}(-2, 2)$ | $\mathcal{U}(-1, 1)$ | $2\pi \cdot \mathcal{U}(0, 1)$ | $2\pi \cdot \mathcal{U}(0.4, 5)$           |
| Hopper-v4      | $\mathcal{U}(-1, 1)$ | $0.0$            | $2\pi \cdot \mathcal{U}(0, 1)$ | $2\pi \cdot \mathcal{U}(0.4, 5)$           |
| Swimmer-v4     | $1.0$            | $0.0$            | $2\pi \cdot \mathcal{U}(0, 1)$ | $2\pi \cdot \mathcal{U}(0.4, 2)$           |
| Walker2d-v4    | $\mathcal{U}(-1, 1)$ | $\mathcal{U}(-1, 1)$ | $2\pi \cdot \mathcal{U}(0, 1)$ | $2\pi \cdot \mathcal{U}(0.4, 6)$           |
| Quadruped      | $\mathcal{U}(-1, 1)$ | $\mathcal{U}(-1, 1)$ | $2\pi \cdot \mathcal{U}(0, 1)$ | $2\pi \cdot \mathcal{U}(0.4, 2)$           |

Table 4: Proportional ($k_p$) and derivative ($k_d$) gains of the PD controller for each environment.

|                | $k_p$ | $k_d$ |
|----------------|-------|-------|
| Ant-v4         | 1.0   | 0.05  |
| HalfCheetah-v4 | 1.0   | 0.05  |
| Hopper-v4      | 10.0  | 0.5   |
| Swimmer-v4     | 7.0   | 0.7   |
| Walker2d-v4    | 10.0  | 0.5   |

## A.3 ABLATION STUDY

In this section, we examine the impact of design choices of Eq. (1) on performance. In particular, we investigate the influence of having phase-dependent frequencies (we set $\omega_{\text{swing}} = \omega_{\text{stance}} = \omega$) and the importance of having phase shifts $\varphi_i$ between oscillators (we set $\varphi_i = 0$). The results are shown in Figs. 7 and 8 and table 5.

The equations of the different variants of Eq. (1) are:

$$
\begin{aligned}
q_i^{\text{des}}(t) &= a_i \cdot \sin(\omega \cdot t + \varphi_i) + b_i & \text{No } \omega_{\text{swing}} \\
q_i^{\text{des}}(t) &= a_i \cdot \sin(\theta_i(t)) + b_i & \text{No } \varphi_i \\
q_i^{\text{des}}(t) &= a_i \cdot \sin(\omega \cdot t) + b_i & \text{No } \varphi_i \text{ No } \omega_{\text{swing}}
\end{aligned}
\tag{2}
$$

where $\theta_i(t)$ is the same as in Eq. (1).

For the HALFCHEETAH, SWIMMER and ANT tasks, having a single frequency $\omega$ is sufficient, while it is critical to have phase-dependent frequencies for the HOPPER environment. The phase shifts $\varphi_i$ are needed when all joints cannot be synchronous (as in the SWIMMER task). For the quadruped, these phase shifts $\varphi_i$ represent the gait and encode symmetries between the legs.

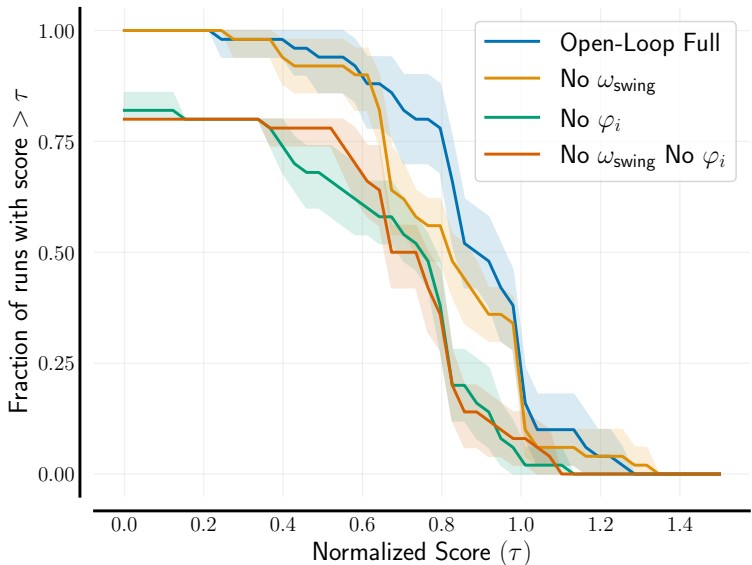

Figure 7: Performance profiles on the MuJoCo locomotion tasks using different variants of the open-loop approach, with a 95% confidence interval.

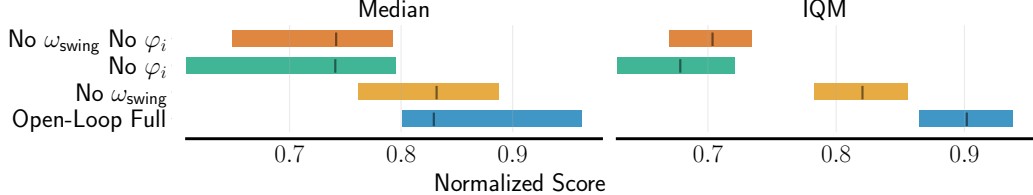

Figure 8: Metrics results on MuJoCo locomotion tasks for the different variants using median and interquartile mean (IQM), with a 95% confidence interval.

Table 5: Results on MuJoCo locomotion tasks (mean and standard error are displayed) with different variant of the approach.

|  | Open-Loop | | | |
|  | No $\varphi_i$ No $\omega_{swing}$ | No $\varphi_i$ | No $\omega_{swing}$ | Full |
|---|---|---|---|---|
| Ant-v4 | 1167 +/- 3 | 1173 +/- 3 | 1239 +/- 8 | 1235 +/- 6 |
| HalfCheetah-v4 | 2221 +/- 27 | 2245 +/- 30 | 2532 +/- 42 | 2400 +/- 31 |
| Hopper-v4 | 929 +/- 9 | 785 +/- 28 | 986 +/- 7 | 1241 +/- 30 |
| Swimmer-v4 | -119 +/- 8 | -82 +/- 6 | 356 +/- 0 | 356 +/- 0 |
| Walker2d-v4 | 1484 +/- 36 | 1482 +/- 34 | 1140 +/- 32 | 1508 +/- 27 |

## A.4 RAW RESULTS ON MUJOCO

Table 6: Results on MuJoCo locomotion tasks (mean and standard error are displayed).

| Environments | SAC | PPO | DDPG | ARS | Open-Loop | |
|---|---|---|---|---|---|---|
| | | | | | 1 x budget | 3 x budget |
| Ant-v4 | 4514 +/- 352 | 796 +/- 116 | 265 +/- 210 | 1241 +/- 25 | 1235 +/- 6 | 2130 +/- 120 |
| HalfCheetah-v4 | 10538 +/- 286 | 1770 +/- 254 | 11267 +/- 317 | 2195 +/- 272 | 2400 +/- 31 | 4003 +/- 100 |
| Hopper-v4 | 4039 +/- 118 | 1817 +/- 312 | 1240 +/- 124 | 2538 +/- 253 | 1241 +/- 30 | 2056 +/- 121 |
| Swimmer-v4 | 240 +/- 39 | 334 +/- 18 | 25 +/- 4 | 267 +/- 31 | 356 +/- 0 | 357 +/- 1 |
| Walker-v4 | 3192 +/- 184 | 1817 +/- 312 | 563 +/- 64 | 444 +/- 10 | 1508 +/- 261 | 2500 +/- 461 |

