# OpenReview forum: "A Simple Open-Loop Baseline for Reinforcement Learning Locomotion Tasks"
_ICLR.cc/2024/Conference — Submitted to ICLR 2024_

### Official Review · Reviewer_bfuq · 2023-10-27

**Soundness:** 3 good
**Presentation:** 3 good
**Contribution:** 2 fair
**Rating:** 5
**Confidence:** 5

**Summary:**

This paper presents a simple baseline for locomotion tasks usually used to evaluate RL algorithms. This simple baseline employs oscillators to generate periodic joint motions, providing a open-loop method with ease of reproducibility, a fracion of the parameters of neural network-based approaches, and little use of computational resources.

**Strengths:**

- One of the paper's notable strengths is the simplicity of the proposed approach. The use of open-loop oscillators to solve locomotion tasks offers an elegant and straightforward solution. This simplicity is in contrast to the increasing complexity of many contemporary deep reinforcement learning (DRL) methods.

- This approach is easily reproducible. It provides a minimal standalone code for solving the swimmer task and includes comprehensive details on the optimization of oscillator parameters.

- I really appreciate the thorough analysis of the proposed approach with a few different RL methods. This analysis provides valuable insights into the performance, efficiency, and robustness of this simple baseline in relation to existing more complex methodologies.

**Weaknesses:**

- Although the method requires minimal parameters, a lot of trial-and-error is needed in determining the number of oscillators to use for each environment, which is in direct contrast to RL methods which are general across multiple environments. The need for fine-tuning in each context may limit the method's scalability.

- While this result is really nice, this is also very expected. When maximizing the default reward of these environments, the resulting policy always corresponds to cyclical behavior which suggests that a simple cyclical controller could solve the task. Furthermore, these environments are very simple, and this work further intesifies the importance of moving on from this simple environments and focus on harder tasks.

- While the simplicity and practicality of the approach are strengths, the contribution is minimal. I see this work as better suited for the blog post track rather than a conference paper.

**Questions:**

No questions. I address my opinions in the fields above. I think this work is somewhat valuable to the community, but I also think that it could easily be more appreciated as a simple blog post as the contribution itself is minimal.

---

> ### Author Response · Authors · 2023-11-14
> **Reply to comments**
>
> Thank you for your comments. We've carefully read your suggestions and would like to provide the following responses.
>
> > Although the method requires minimal parameters, a lot of trial-and-error is needed in determining the number of oscillators to use for each environment, which is in direct contrast to RL methods which are general across multiple environments. The need for fine-tuning in each context may limit the method's scalability.
>
> Thank you for your remark.
> It seems that our initial description may have led to a misunderstanding.
> To clarify, the number of oscillators used in our method is the same as the number of joints, since they operate in joint space.
> In practice, extending the method to additional environments is relatively straightforward; for example, it took the authors only 20 minutes to modify the code from Hopper to Walker.
>
>
> > While this result is really nice, this is also very expected. When maximizing the default reward of these environments, the resulting policy always corresponds to cyclical behavior which suggests that a simple cyclical controller could solve the task. Furthermore, these environments are very simple, and this work further intesifies the importance of moving on from this simple environments and focus on harder tasks.
>
> While the success of the open-loop oscillators in the Swimmer environment is anticipated, its effectiveness in the Walker, Hopper or quadruped environments is more unexpected, as one might assume that feedback control or inverse kinematics would be necessary to balance the robots or to learn a meaningful open-loop policy.
> Another surprising fact is the tiny number of parameters needed compared to DRL algorithms like DDPG, which have almost a million of parameters.
> The results with RBF policies or ARS with linear policy were already unexpected when they came out [1, 2].
>
> It is true that studies of nature and experiments have shown that periodic control is at the heart of locomotion [3], but it is unclear and surprising how simple a controller can be.
> As discussed in the last section of the paper, the fact that a single global frequency (per phase for hopper/walker) shared by each joint oscillator is sufficient is also an interesting finding related to another field that studies the intrinsic oscillation of robotic systems (nonlinear modes).
>
> The surprise expressed by R2:vPWP and R3:w6U7 at the failure of DRL in a seemingly straightforward sim-to-real transfer scenario underscores the importance of the insights presented in our paper.
> By challenging common assumptions and exploring alternative solutions, we hope to contribute to the application of RL to robotics.
>
> We updated the discussion section to underscore more the unexpected results from our paper.
>
> [1] Rajeswaran, A. et al, Neurips 2017. Towards generalization and simplicity in continuous control. \
> [2] Mania, H. et al. Neurips 2018. Simple random search of static linear policies is competitive for reinforcement learning.
> [3] Ijspeert, A (2008) "Central pattern generators for locomotion control in animals and robots: a review."
>
>
> > Furthermore, these environments are very simple, and this work further intesifies the importance of moving on from this simple environments and focus on harder tasks.
>
> MuJoCo environments are indeed simple, but have been the basis for advances in continuous control for RL, and are still widely used (for both online and offline RL).
> Our intention is to raise awareness of the failures of DRL on simple problems like the elastic quadruped, and to open discussion on the cost of complexity/generality, and why many heuristics are needed for simulation-to-reality scenarios.
>
> We would also like to point out that although the task is simple for the elastic quadruped, it is not a trivial environment.
> The serial elastic actuators are more challenging to control than their rigid counterparts because they introduce more complex dynamics.
>
> Results on more complex environment would make it harder to understand the failures and role of each component, which would go against recommendation by [4].
>
> We updated the discussion section to highlight why results on simple environment can still provide valuable insights for the RL community.
>
>
> [4] Patterson, A, et al. (2023) "Empirical Design in Reinforcement Learning."

---

> > ### Comment · Reviewer_bfuq · 2023-11-20
> > **Thanks for your comment.**
> >
> > Thanks for the detailed comments. I don't have any more questions. Nevertheless, I think the current score still matches the contribution and importance of the work for the community.

---

### Official Review · Reviewer_w6U7 · 2023-10-29

**Soundness:** 2 fair
**Presentation:** 2 fair
**Contribution:** 1 poor
**Rating:** 3
**Confidence:** 4

**Summary:**

The presented paper introduces a Central Pattern Generators (CPG) controller as a rudimentary open-loop baseline for reinforcement learning (RL) in locomotion tasks. In addition to detailing the method, the authors have exhaustively evaluated their proposed baseline on Mujoco-v4 environments and deployed it in a simple real-world setting.

**Strengths:**

1. The paper offers a comprehensible open-loop RL baseline, with the technical aspects meticulously described and supplementary implementation details provided in the appendix.
2. The research does not limit itself to simulation; it extends its evaluation of the open-loop controller to a genuine real-world environment.
3. The authors emphasize the significance of leveraging prior knowledge in task formulation, suggesting its equal importance to algorithmic enhancement.
4. Common pitfalls of existing RL techniques are explored, accompanied by results that demonstrate scenarios in which contemporary RL methods are outperformed by the simple open-loop baseline.

**Weaknesses:**

1. In terms of novelty of the work, central pattern generators have been applied to legged locomotion in different ways in much more realistic and complex setting, for example it’s been used to build a better action space for RL as described in Bellegarda, et al [1].  In this case, it’s hard to see the value of applying CPG to many simple locomotion tasks (Mujoco-v4 in gymnasium). Though Gym contains simple locomotion tasks, no matter whether it’s hard to solve or not, it’s mainly designed for evaluating different RL methods providing a specific manually designed controller for these tasks can hardly address issues in current RL methods.
2. Though this work is compared with several RL methods trying to study the robustness of current RL methods, no recent works are compared. The most recent work compared in this work is SAC.
3. Though the proposed simple open-loop baseline controller is robust and provide good sample efficiency, this open-loop baseline can only be applied to locomotion tasks, where current RL method can solve much more complex locomotion tasks (like traverse complex terrain) within one hour [2, 3].

Reference:
[1] Bellegarda, et al. CPG-RL: Learning central pattern generators for quadruped loco- motion.

[2] Rudin, et al. Learning to Walk in Minutes Using Massively Parallel Deep Reinforcement Learning

[3] Smith, et al. A Walk in the Park: Learning to Walk in 20 Minutes With Model-Free Reinforcement Learning

**Questions:**

1. It would be great if authors could further address the contribution of the proposed work.
2. It would be great if this work can provide more comparison with recent RL works if this work is trying to address the issue of existing RL methods, like AWR [1], V-MPO[2] (just example), or some model-based RL methods[3].
3. When it comes to sim2real, many rewards like minimizing the energy consumption, penalize the high frequency action has been well-studied, as well as the domain randomization techniques, when studying sim2real performance, these techniques should be applied to see the actual performance different. It would be great to apply these techniques if this work is trying to claim the performance of proposed method in sim2real transfer.

Reference:
[1] Peng, et al. Advantage-Weighted Regression: Simple and Scalable Off-Policy Reinforcement Learning

[2] Song, et al. V-MPO: On-Policy Maximum a Posteriori Policy Optimization for Discrete and Continuous Control

[3] Hansen, et al. Temporal Difference Learning for Model Predictive Control

---

> ### Author Response · Authors · 2023-11-14
> **Reply to comments**
>
> Thanks for your feedback. The following are our replies to the questions, and we have updated the paper accordingly.
>
> > central pattern generators have been applied to legged locomotion in different ways in much more realistic and complex setting
> > In this case, it’s hard to see the value of applying CPG to many simple locomotion tasks (Mujoco-v4 in gymnasium). Though Gym contains simple locomotion tasks, no matter whether it’s hard to solve or not, it’s mainly designed for evaluating different RL methods providing a specific manually designed controller for these tasks can hardly address issues in current RL methods.
>
> Thanks for your comment.
>
> Our goal is not to replace DRL methods, nor to claim to be state of the art, but as the reviewer noted, to provide insight into "common pitfalls of existing RL techniques" (see Sections 4.3 and 4.4 for instance), as recommended in section 4 of [1].
> The MuJoCo locomotion tasks are indeed simple, but they still require hundreds of thousands of parameters and thousands of trials to solve, and are widely used in the RL community.
> The complexity of DRL algorithm makes it hard to reproduce, and the computation power required prevent them from being used on an on-board computer.
> Being aware of the failures of DRL on simple problems like the elastic quadruped, which illustrates the challenges of sim2real, and understanding the cost of complexity is crucial for improving application of RL to robotics.
>
> To clarify the position of the paper, we have updated the abstract, the introduction and discussion sections.
>
> [1] Patterson, A, et al. (2023) "Empirical Design in Reinforcement Learning."
>
>
> > Though this work is compared with several RL methods trying to study the robustness of current RL methods, no recent works are compared. The most recent work compared in this work is SAC.
> > It would be great if this work can provide more comparison with recent RL works if this work is trying to address the issue of existing RL methods, like AWR, V-MPO (just example), or some model-based RL methods.
>
> Thank you for your remark.
>
> We use SAC v3 (with TD3 double critics and automatic entropy tuning [2]), which is an algorithm that provides strong baselines for MuJoCo tasks (similar performance to AWR), is widely used for continuous control problems, and is the basis for recent sample-efficient variants like TQC, REDQ or DroQ [3, 4, 5].
>
> We could replace SAC with its distributional variant TQC [5], but this won't change the findings or the discussion of the paper.
> TQC has slightly better performance in simulation (still much better than a simple open-loop baseline) and a worse runtime compared to SAC [6].
> However, because it shares the same policy structure (among other elements), it has the same shortcomings as SAC for the sim2real experiment (e.g., learning high-frequency control), and that's exactly why we use SAC as a baseline.
> We believe that SAC is already complex enough (especially with respect to an open-loop model-free baseline), and since SAC and its variants are widely used by the RL community in robotics, providing insight into SAC's pitfalls provides insight into the variants that share the same components.
>
> It is for a similar reason that we use DDPG, to provide better context for the reader (following [1]), despite it being outdated.
>
> We updated the experiments' section to reflect our choice.
>
> [1] Patterson, A, et al. (2023) "Empirical Design in Reinforcement Learning." \
> [2] Haarnoja, T, et al. (2019) "Learning to walk via deep reinforcement learning." \
> [3] Chen, X, et al. (2020) "Randomized ensembled double q-learning: Learning fast without a model." \
> [4] Hiraoka, T, et al. (2022) "Dropout q-functions for doubly efficient reinforcement learning." \
> [5] Kuznetsov, A, et al. (2020) "Controlling overestimation bias with truncated mixture of continuous distributional quantile critics." \
> [6] https://github.com/openrlbenchmark/openrlbenchmark and https://wandb.ai/openrlbenchmark/sb3
>
>
> > It would be great to apply these techniques if this work is trying to claim the performance of proposed method in sim2real transfer.
>
> Thank you for your comment.
>
> We intentionally simplified the environment and removed task-specific heuristics in our sim2real experiments to understand why RL fails in this case and to observe the resulting behavior.
> Our goal is to provide insight and stimulate discussion about the trade-offs between complexity, generality, and performance in current deep RL algorithms.
>
> While we could improve our results by adding complex reward engineering, randomization, or low-pass filtering, our intention is to propose the simplest model-free baseline possible.
> It would also be unfair to compare our simple open-loop baseline to a highly tuned DRL algorithm that incorporates many implementation details, uses the state as input, and has 10,000 more parameters.
>
> We have updated the sim2real section to make that point clearer.

---

### Official Review · Reviewer_vPWP · 2023-10-29

**Soundness:** 2 fair
**Presentation:** 3 good
**Contribution:** 1 poor
**Rating:** 3
**Confidence:** 4

**Summary:**

The paper proposes an open-loop control algorithm to solve five different locomotion control tasks. The experiments and video show that a simple control algorithm can achieve better results than RL policy.

**Strengths:**

- The paper contains the real robot experiments. It shows the real effectiveness of the proposed algorithm.
- The paper considers the advantages of the method from the perspective of runtime.

**Weaknesses:**

- The method is not innovative. In fact, the baseline of many previous RL work was a simple open-loop controller. However, whether those works only used RL [1] or considered the combination of RL and classical control algorithms [2][3], they all produced much more impressive locomotion control results.
- I think the training results of RL methods in this paper are much worse than the level of other RL for locomotion papers, and they have not considered clearly what problems RL for control policy solves (like robustness in unseen environments).

- The paper has limited inspiration for the field of machine learning, which is the main topic of this conference.

[1] Ashish Kumar, Zipeng Fu, Deepak Pathak, Jitendra Malik, "RMA: Rapid Motor Adaptation for Legged Robots"

[2] Atil Iscen, Ken Caluwaerts, Jie Tan, Tingnan Zhang, Erwin Coumans, Vikas Sindhwani, Vincent Vanhoucke, "Policies Modulating Trajectory Generators"

[3] Takahiro Miki, Joonho Lee, Jemin Hwangbo, Lorenz Wellhausen, Vladlen Koltun, Marco Hutter, "Learning robust perceptive locomotion for quadrupedal robots in the wild"

**Questions:**

- In experiments, why the authers choose speed as the metric of the locomotion task? Can you consider more realistic metrics like stability under the noises or some uneven terrains, as well as show more videos beyond walking in a straight line?
- The method works because the tasks only depend on cyclic movements for joints. Can the proposed method still be used if there are more obstacles in the environment, which need some complex behavior (turn left/right)?
- Is it possible to try more to adjust the parameters of the pd controller and the range of the joint target position generated by the RL policy to improve the results of the RL method? The demo shown by the authors is indeed worse than other RL-based locomotion works recently.

---

> ### Author Response · Authors · 2023-11-14
> **Reply to comments part 1**
>
> Thank you for your remarks. We've carefully considered your suggestions and would like to provide the following responses.
>
> > The method is not innovative. In fact, the baseline of many previous RL work was a simple open-loop controller. However, whether those works only used RL [1] or considered the combination of RL and classical control algorithms [2][3], they all produced much more impressive locomotion control results.
>
> Thank you for your feedback.
>
> The authors assume that the reviewer is referring to Kumar et al. (2021), Iscen et al. (2019) and Miki et al. (2022) when mentioning that "the baseline of many previous RL work was a simple open-loop controller".
> (If we are mistaken, we kindly ask the reviewer to provide us with specific references. We will gladly incorporate them into our related work section.)
>
> We believe that our work has a different scope than these three papers.
>
> In Kumar et al. (2021), the focus is on proposing a new method for sim2real transfer, not understanding RL failures through a comparison with an open-loop baseline, which we were not able to find in that work.
> Iscen et al. (2019) propose a new approach to combine learning and trajectory generation. The trajectory generator is tailored to the quadruped problem (defined in task space) and is not compared to the learned policy (only briefly mentioned in the experiment section).
> In Miki et al. (2022), the goal is to learn robust locomotion policies using a height scan in addition to proprioceptive data.
> The observation and action spaces are inspired by CPG, but no open-loop baseline is used.
>
> In summary, the three papers do not propose a simple baseline for RL tasks but rather focus on bootstrapping learning with robot knowledge, and provide results only for quadrupeds.
> In contrast, we present a simple baseline that is easy to study, easy to adapt to new problems and evaluate it on a popular benchmark using six different robot morphologies.
>
> We would also like to clarify that the MuJoCo locomotion tasks, which we use in our experiments, were originally designed to evaluate model-based control methods [1], and were later adopted for continuous control RL.
> To the best of our knowledge, the simplest baseline previously employed for these tasks was a linear policy [2], while open-loop control has not been extensively explored in this context.
>
> Since the MuJoCo locomotion tasks are widely used in the RL community (for both online and offline RL), we believe that providing a very simple baseline would provide valuable insights and better understanding of RL algorithms and their limitations (see Section 4.3 for instance).
>
> [1] Todorov E. et al. (2012) "Mujoco: A physics engine for model-based control." \
> [2] Mania H. et al. (2018) "Simple random search provides a competitive approach to reinforcement learning"

---

> ### Author Response · Authors · 2023-11-14
> **Reply to comments part 2**
>
> > However, whether those works only used RL or considered the combination of RL and classical control algorithms, they all produced much more impressive locomotion control results.
> >  [...] show more videos beyond walking in a straight line?
>
> Thank you for your remark.
> It appears we had not yet made the paper’s stance clear enough.
> Our intention is not to replace DRL algorithms or to advance SOTA in performance, but to shed light on their limitations (such as failure on simple problems [3, 4, 5]) and to open discussion on their current complexity and "common pitfalls" (as noted by R1:nvZX, R3:w6U7, and R4:bfuq).
>
> Following the recommendations of Section 4 of [6], we designed the experiments to provide "insights rather than state-of-the-art claims".
> For example, we show that only ten parameters are sufficient for a model-free approach that doesn't depend on the current state in several locomotion tasks, compared to the hundreds of thousands used by current DRL algorithms.
> Similarly, our results suggest that DRL algorithms are very brittle when exposed to sensor noise or failure, even on an uninformative sensor (this was confirmed by the experiment we added where SAC is trained with noise on the first sensor).
>
> Current DRL algorithms can indeed solve more challenging locomotion tasks, but despite their generality, they require numerous implementation details and careful task design such as reward engineering, low-pass filtering or domain randomization (mentioned in the "discussion" section).
> Without these heuristics, DRL succeeds in simulation but fails to transfer on simple tasks like the one presented in the paper.
> On the same task, a very simple baseline that uses a single frequency for all joints succeeds.
> We believe that understanding the failure case on simple problems that are commonly used in the RL community is important for advancing the field.
>
> To clarify the position of the paper, we have updated the abstract, the introduction and discussion sections.
> If the reviewer has other suggestions to make our intentions clearer, we would be happy to incorporate them.
>
> [3] Colas, C et al. (2018) "Gep-pg: Decoupling exploration and exploitation in deep reinforcement learning algorithms." \
> [4] Franceschetti, Maël, et al. (2022) "Making Reinforcement Learning Work on Swimmer." \
> [5] Matheron, G et al. (2019) "The problem with DDPG: understanding failures in deterministic environments with sparse rewards." \
> [6] Patterson, A, et al. (2023) "Empirical Design in Reinforcement Learning."
>
>
> > The paper has limited inspiration for the field of machine learning, which is the main topic of this conference.
>
> We acknowledge that our paper provides a robotics perspective for reinforcement learning practitioners.
> However, we also believe that the insights provided in the paper (and demonstrated by the real robot experiment) are valuable for applications of RL in robotics (which is one of the topics mentioned in the ICLR 2024 paper call [7], see "applications to robotics, autonomy, planning").
>
> [7] https://iclr.cc/Conferences/2024/CallForPapers
>
>
> > why the authers choose speed as the metric of the locomotion task?
>
> We chose this metric because it was the simplest to define the task (only one reward term) and we could compare it to other controllers developed for the elastic quadruped robot.
>
> > Can the proposed method still be used if there are more obstacles in the environment, which need some complex behavior (turn left/right)?
>
> As mentioned in the "limitation" section of the paper and in the reply above, open-loop control cannot, by design, address tasks that require responding to variations in the environment.
> However, the aim of the paper is not to propose a method that can replace RL, but provide a point of reference that is easy to study and that doesn't fail where "vanilla" DRL can be brittle (e.g. with sim2real transfer or when exposed to sensor noise).

---

> > ### Comment · Reviewer_vPWP · 2023-11-20
> > **Reply to Authors**
> >
> > Thank the authors for the detailed comments. Some of my questions have been addressed. However, because the given results of the RL baseline are truly worse than the previous DRL for locomotion papers as I listed, it is unfair to raise the score or believe the proposed method is insightful enough for the area.

---

> ### Author Response · Authors · 2023-11-14
> **Reply to comments part 3**
>
> > Is it possible to try more to adjust the parameters of the pd controller and the range of the joint target position generated by the RL policy to improve the results of the RL method? The demo shown by the authors is indeed worse than other RL-based locomotion works recently
>
> Thank you for your remark.
> The result of the RL controller on the real robot is indeed worse than the recently published RL based locomotion, because we intentionally removed all the task specific heuristics and tried to make the environment as simple as possible.
> This is one of the main points of the paper, which is to understand why RL fails in this case and what is the observed behavior.
>
> We could improve the results by adding complex reward engineering, randomization, or low-pass filtering (or equivalently tuning the PD controller until it works at the cost of slowness).
> However, our intention is to provide insight and food for thought about the cost of complexity and generality of the current DRL algorithm by proposing the simplest model-free baseline possible.
> It would also be unfair to compare the simple open-loop baseline to a highly tuned DRL algorithm, that includes already many implementation details, uses the state as an input and has 10 000 more parameters.
>
> We have updated the sim2real section to make that point clearer.

---

### Official Review · Reviewer_nvZX · 2023-10-30

**Soundness:** 4 excellent
**Presentation:** 4 excellent
**Contribution:** 3 good
**Rating:** 6
**Confidence:** 4

**Summary:**

This paper proposes an open-loop controller baseline that has decent performance on a number of control tasks. The baseline is shown to be robust to noise and can be transferred to a real robot.

**Strengths:**

The primary strength of the approach is in its simplicity. The authors are careful to argue that the proposed method achieves ‘satisfactory’ performance on a number of tasks without requiring complex models. As the authors point out, there are natural advantages to using simpler pattern generators for robotics which avoid issues of bang-bang control and wear-and-tear that might arise from learnt methods. I also appreciate that the authors are careful with their claims and acknowledge that SAC outperforms their proposed approach in simulation without noise.

The paper is also written in clear and simple language and is easy to follow.

**Weaknesses:**

I generally support the argument for simplicity that is presented in the paper. The baseline requires far fewer parameters and less computation to train. However, I think the paper is missing a discussion on how RL might still play a role when applied to robotics.

There are solutions to the issues presented in Section 4.3 for Robustness to sensor noise that RL practitioners would likely implement. For example, noise can be added in simulation during the RL training which would result in a conservative but more robust policy. It would be interesting to see how that compares to an open loop baseline. As the paper argues, domain specific knowledge could help improve algorithm design albeit for the RL algorithms in this case. More generally, the results seem to indicate that if final performance is the key driver being optimised, RL may still be the tool of choice.

Finally, while the paper does a good job of implementing baselines from RL and evolutionary algorithms, there are no pattern generating baselines being compared to. Central pattern generators have been studied for some time so it would be important to know how the proposed open-loop baseline compares against existing ideas in the field.

**Questions:**

1. The `Contributions` list that the baseline can handle sparse rewards. As far as I know none of the environments used for the evaluation use sparse rewards though. Could the authors clarify this point?
2. Can the authors include another non-learning baseline to understand how much is gained with the specific implementation proposed?
3. For the robustness to sensor noise, can the authors include a baseline where SAC is trained with a subset of the noisy parameters? For instance, if SAC were trained to withstand random noise of say 3N, it seems more likely that it would be able to withstand the 5N noise tested in the paper.
4. For Figure 3, shouldn’t the dot for `Open Loop` sit exactly at 1? It seems that the plot is slightly below 1 which seems off to me.
5. While I like the general evaluation in the Experiments section, I found Table 6 in the Appendix with the actual scores informative. Since there is still space, I would suggest adding that table to the main text - if the authors think it would not affect the narrative flow too much.

---

> ### Author Response · Authors · 2023-11-14
> **Reply to comments part 1**
>
> Thank you for your time and valuable suggestions!
> Please find below our replies to your concerns and suggestions.
>
> > However, I think the paper is missing a discussion on how RL might still play a role when applied to robotics.
>
> We agree that RL has a role to play in robotics (especially for locomotion tasks).
> Our intention was to convey this in the earlier version.
> We have updated the discussion section to make this statement clearer.
>
> > More generally, the results seem to indicate that if final performance is the key driver being optimised, RL may still be the tool of choice.
>
> RL is the tool of choice for achieving top performance in simulation, as demonstrated by our experiment and discussed in the paper (see Table 2).
> However, applying RL in real-world scenarios requires numerous additional techniques, such as reward engineering, low-pass filtering, domain randomization, and leveraging privileged information.
>
> Our goal is not to replace DRL algorithms but to highlight their limitations, including their struggles with simple problems.
> In doing so, we aim to raise awareness within the community about the tradeoffs between the generality of these algorithms and the effectiveness of a much simpler baseline, which has "natural advantages" as noted by the reviewer.
>
> To clarify this point, we have updated the abstract, the introduction and discussion sections.
>
>
> > there are no pattern generating baselines being compared to.
> > Can the authors include another non-learning baseline to understand how much is gained with the specific implementation proposed?
>
> The authors interpret the reviewer's suggestion as an endorsement for the inclusion of a CPG baseline.
>
> Our proposed baseline can be seen as a simplified version of [1, 2], which has similar properties (smooth by design, open-loop, periodic), but is much easier to implement and adapt to new problems.
> In fact, the CPG presented in these works was our starting point, but it was tailored for quadruped locomotion (using IK and fixed coupling matrix).
> To be able to adapt it to all MuJoCo locomotion tasks, we simplified it, removed the constraint of a fixed coupling and converged to the simple oscillators presented in the paper.
>
> Additionally, our intention is to provide the simplest baseline for RL locomotion tasks and provide insights for RL algorithms (following [3]), not to advance SOTA in the CPG domain.
>
> We updated the introduction and changed the order of the related work to emphasize the main message of the paper.
>
> [1] Bellegarda, G., & Ijspeert, A. (2022). CPG-RL: Learning central pattern generators for quadruped locomotion. \
> [2] Raffin, A, et al. (2022) "Learning to Exploit Elastic Actuators for Quadruped Locomotion." \
> [3] Patterson, A, et al. (2023) "Empirical Design in Reinforcement Learning."
>
>
> > The Contributions list that the baseline can handle sparse rewards. As far as I know none of the environments used for the evaluation use sparse rewards though. Could the authors clarify this point?
>
> Yes, the baseline we propose can handle sparse rewards, which is a benefit of black-box optimization (BBO).
> Because BBO uses only episodic returns, rather than immediate rewards, the baseline and ARS can effectively tackle problems with sparse or delayed rewards.
> Since this is a known pitfall of DRL algorithms, we wanted to mention this advantage without making it the main focus of our work.
> For this reason we did not focus on this aspect during the evaluation but chose to highlight it textually instead.
>
> We added a sentence when presenting the method to make that point clearer.
>
>
> > For the robustness to sensor noise, can the authors include a baseline where SAC is trained with a subset of the noisy parameters? For instance, if SAC were trained to withstand random noise of say 3N, it seems more likely that it would be able to withstand the 5N noise tested in the paper.
>
> Thanks for the suggestion, we added an experiment where SAC is trained with a noisy sensor to the paper.
>
>
> > For Figure 3, shouldn’t the dot for Open Loop sit exactly at 1? It seems that the plot is slightly below 1 which seems off to me.
>
> The plot is indeed slightly below one.
> This is because we normalize with respect to the maximum performance, and since there is some variance between runs (see performance profiles), the normalized performance of the oscillators is not exactly one, but close to one.

---

> ### Author Response · Authors · 2023-11-14
> **Reply to comments part 2**
>
> > While I like the general evaluation in the Experiments section, I found Table 6 in the Appendix with the actual scores informative. Since there is still space, I would suggest adding that table to the main text - if the authors think it would not affect the narrative flow too much.
>
> This is indeed a conscious choice.
> The authors follow the best practice recommendations from [4 - NeurIPS 21 outstanding paper, 5] and provide a better statistical representation of performance through the use of IQM, stratified bootstrap, performance profiles, and probability of improvement.
>
> We would like to emphasize that while performance is a critical aspect, runtime and parameter efficiency are another.
> To maintain the clarity of the paper's narrative, we have chosen not to include the table in the main text.
> Instead, it is provided in the appendix for readers who are interested in exploring this aspect further.
>
> [4] Agarwal, R, et al. (2021) "Deep reinforcement learning at the edge of the statistical precipice." \
> [5] Patterson, A, et al. (2023) "Empirical Design in Reinforcement Learning."

---

> > ### Comment · Reviewer_nvZX · 2023-11-21
> > **Response to authors**
> >
> > Thank you for your responses and adding the additional experiment of SAC with noise. I appreciate the simplicity of the approach and think it is valuable to understand the pitfalls of RL for control. I will leave my score as is for now.

---

### Author Response · Authors · 2023-11-14
**Reply to all reviewers**

The authors appreciate the reviewers' constructive feedback, which undoubtedly enhanced the paper's quality.

As noted by reviewers, we propose an "elegant and straightforward solution" for locomotion tasks (R1:nvZX, R4:bfuq), which is "easily reproducible" (R3:w6U7, R4:bfuq).
This approach capitalizes on the "natural advantages" of open-loop oscillators, particularly in "real robot experiments" (R1:nvZX, R2:vPWP, R3:w6U7, R4:bfuq).
Our intention behind introducing this baseline isn't to replace RL but rather to investigate "common pitfalls of existing RL techniques" (R1:nvZX) and "provide valuable insights into the performance, efficiency, and robustness of this simple baseline in relation to existing more complex methodologies" (R3:w6U7, R4:bfuq).


## Summary of changes
- We added an experiment where SAC is trained with a noisy sensor
- We updated the abstract, introduction and discussion to clarify the aims of the paper
- We changed the order of the related work to emphasize the main message of the paper
- We updated the method section to clarify why the approach can handle sparse rewards
- We updated the experiment section to explain why SAC was chosen as a baseline
- We added a paragraph in the discussion section about the unexpected results from our study
- We updated the sim2real section to clarify why we train without randomization or reward engineering

EDIT: we provide a diff version of the pdf here: https://pdfhost.io/v/F7MKOrTfO_open_loop_diff

---

### Author Response · Authors · 2023-11-20
**About our answers**

Dear Reviewers,

The discussion phase between the authors and the reviewers will soon come to an end. We have received no further comments or questions. May we assume that all your concerns or questions have been addressed? If so, could you please consider adjusting the recommendation score? Thank you very much!

---

### Meta-Review · Area_Chair_as7B · 2023-12-24

**Metareview:**

### Summary
The paper presents a baseline for locomotion tasks usually used to evaluate RL algorithms.
The core contribution is provide a simple baseline for locomotion tasks, with the aim of highlighting the current shortcomings and compromises of DeepRL algorithms, even in seemingly simple tasks
This baseline employs oscillators to generate periodic joint motions, providing a open-loop method with ease of reproducibility, a fraction of the parameters of neural network-based approaches, and little use of computational resources.

###  Strengths
+ Simplicity of approach and hence reproducibility,
+ Strong Experimental evaluation

### Weaknesses:
- Limited extensibility due to the need to fit to each environment.
- Limited to only cyclical tasks, and also application of known results in CPG.
- Comparison to RL, which is more generically applicable to more than cyclical gait patterns, would not be on equal terms.
	- It would help to submit this method with a set of prebuild environments as a benchmark. Furthermore comparison should be provided with more recent results in RL using large scale simulation and randomization.

**Justification For Why Not Higher Score:**

Reviewers agreed on retaining the scores after discussion, and the AC agrees on limited contributions.

**Justification For Why Not Lower Score:**

All Reviewers are in agreement of the simplicity and yet competitive baseline methods in locomotion.

---

### Decision · Program_Chairs · 2024-01-16

Reject